# Concise Overview of Methodologies Employed in the Study of Bacterial DNA Replication

**DOI:** 10.3390/ijms26020446

**Published:** 2025-01-07

**Authors:** Monika Maciąg-Dorszyńska, Joanna Morcinek-Orłowska, Sylwia Barańska

**Affiliations:** 1Department of Bacterial Molecular Genetics, University of Gdansk, Wita Stwosza 59, 80-308 Gdansk, Poland; 2Structural Biology Laboratory, Intercollegiate Faculty of Biotechnology, University of Gdansk and Medical University of Gdansk, Abrahama 58, 80-307 Gdansk, Poland; joanna.morcinek-orlowska@ug.edu.pl

**Keywords:** DNA replication, methods, techniques, bacteria, genetic engineering, microscopy, electrophoresis, flow cytometry

## Abstract

DNA replication is a fundamental process in the cell on which the functioning of the entire cell as well as the maintenance of the entire species depends. This process is synchronized with all other processes within the cell as well as with external, environmental factors. This complex network of interconnections presents significant challenges in the field of DNA replication research, both in terms of identifying an appropriate approach to a question posed and in terms of methodology. This article aims to provide a roadmap to assist in navigating (to help overcome) these challenges and in selecting an appropriate research methodology. It should help to establish a research pathway, starting with arranging the host genetic background for analysis at different cellular levels, which can be achieved using complex or simple single-purpose techniques.

## 1. Introduction

DNA replication is a fundamental process occurring in all living cells. The fidelity of genetic material duplication, maintenance of its appropriate structure, and its segregation into progeny cells are essential for the stable functioning of the entire cell, as well as the organism as a whole. This process provides a crucial basis for the expression of all genes in the cell. Moreover, a significant number of proteins engaged in DNA replication is closely associated with other critical processes within the same cell, including metabolism, cell development, cell division, and the segregation of genetic material into daughter cells [1]. The precision of this process is therefore fundamental to the success of all other processes occurring within the cell. An understanding of DNA replication and its regulation allows us to comprehend mechanisms related to genome evolution, adaptation to changing environments, virulence, and many others. Consequently, it provides the basis for the genetic manipulation of microorganisms, thereby facilitating their application in the sectors of medicine, pharmacology, biotechnology, or agriculture. It also allows us to understand the changes that microorganisms undergo through evolution or experimental pressure, which may not always match our initial expectations. The intricacy of the replication process and the interdependence of the factors involved in other pertinent cellular processes, both give rise to the complexity of experimental design and also necessitate the use of diverse methodologies, frequently focused on a range of cellular levels, from macro- to micro-techniques. Although the process of DNA replication is generally the same for all cell types, there is a considerable variation between organisms in terms of the specific steps involved or the manner of its regulation. It should be noted that not all methods are applicable to the study of both prokaryotes and eukaryotes. Also, the experimental procedures differ significantly between these organisms. Similarly, an alternative experimental methodology is required for the study of viruses and bacteriophages.

Selecting a proper experimental strategy is of crucial importance to test a given research hypothesis and achieve a given goal. It not only includes the choice of the ’core’ experimental methods, where the replication parameters and their changes are examined, but also the establishment of the experimental layout before the pivotal research step. Planning one’s research needs to be preceded by the consideration of the target addressing DNA replication changes. Is it the DNA molecule itself, the proteins involved in the DNA replication process, or the bacterial cell taken at a single- or population-based level? The answer to this question affects the choice of methodology, as well as the decision of whether to study DNA replication in vivo or in vitro. From our own experience, we are aware that planning one’s work is considerably more efficient when the methodological basis, the potential offered by the methods in question, and the knowledge that can be gained from them are known. The ability to select an appropriate methodology enables a reduction in the research time, the ability of obtaining pertinent information in a relatively short time frame, and the opening of options for further research and the modification or implementation of the microorganism in application studies. This article focuses exclusively on bacterial research, integrating both the authors’ expertise and the insights gained from pursuing optimal solutions in our investigations. We hope that it will become a roadmap of methods for the study of DNA replication that the reader can browse through to select both an appropriate scheme and methods according to their facilities and questions posed.

## 2. Key Cellular Processes

### 2.1. Cell Cycle in Prokaryotes

Prokaryotic cells, although known for their relative simplicity, undergo a coordinated growth and division cycle to maintain and propagate their genetic material. The processes of DNA replication and cell division in these single-celled organisms are tightly regulated to ensure the precise duplication and distribution of genetic material [2,3]. The bacterial cellular life cycle consists of three key stages: Phase B, which spans from the cell’s birth to the beginning of replication; Phase C, during which DNA replication takes place; and Phase D, which covers the period between the completion of replication and cell division [2,4] (Figure 1A, upper panel). In *Escherichia coli*, DNA replication commences at a particular chromosomal site referred to as the “*origin* of replication” (*oriC*). This process occurs bidirectionally and ends at the termination region (*ter*). Such a standard cell cycle pattern occurs when bacteria grow in nutrient-poor environments. Studies on the Gram-negative bacterium *Escherichia coli* and the Gram-positive *Bacillus subtilis* have demonstrated that, even under nutrient-rich conditions, the duration of both the C-period and D-period remain essentially constant, although less time is required for cell doubling (Figure 1A,B).

As growth rates increase, both *E. coli* and *B. subtilis* encounter a challenge where the time necessary for chromosome replication and segregation surpasses the time needed to double their mass. To resolve this issue, bacteria resort to a phenomenon called ‘multifork’ replication, first explained by Cooper and Helmstetter in 1968. The authors proposed the initiation of a subsequent round of replication before finishing the previous one. As a consequence, cells may harbor multiple copies of *oriC*, resulting in an *origin*-to-*terminus* ratio of 2*n*:1, where ‘n’ indicates the number of initiated replication rounds [5] (Figure 1B,C). Subsequently, after DNA replication and genetic material segregation, the cell divides. Cellular division is a crucial process that ensures the reproduction and growth of all living organisms. In prokaryotic cells, cell division is a pivotal event in their life cycle. Unlike eukaryotic cells, which experience mitosis or meiosis, prokaryotic cells utilize a more straightforward process for cellular division, known as binary fission. Binary fission enables a single prokaryotic cell to divide into two genetically identical daughter cells, thus promoting the propagation of the species. This highly regulated process is crucial in maintaining genetic stability and survival under changing environmental conditions.

### 2.2. DNA Replication

In bacterial cells, the DNA replication proceeds bidirectionally from *oriC* to the *ter* site. The synthesis of DNA is intricately controlled by mechanisms that precisely dictate the spatial and temporal assembly of new replication forks. It is generally accepted to divide the process of DNA replication into three basic stages: initiation, i.e., the events leading up to the beginning of novel DNA strand synthesis; elongation, comprising the events involved in elongating the DNA chain; and termination, i.e., all the events leading up to the completion of replication. Defining precise boundaries between the various stages of DNA replication is challenging due to cross-boundary processes and regulations. It is assumed that the replication initiation phase encompasses the exit of the multi-protein replication complex from the *origin* site and incorporation of the first nucleotides, as well as all the events immediately preceding this stage. It is also assumed that the recognition of the *origin* sequence, dissociation of the double-stranded DNA structure at the *ori* site, and reconstruction of the multi-protein complex known as the replisome are the necessary events required to initiate the synthesis of new DNA strands. Going further, the replisome displays various functions stemming from its different components’ contribution. These include DNA unwinding by the helicase (*E. coli* DnaB), the synthesis of short primers by the primase or polymerase (e.g., *E. coli* DnaG primase), and the complex’s processivity provided by the sliding ring clamps (*E. coli* DnaN) and their associated loaders (*E. coli* DnaE, DnaQ, HolE), as well as the binding of protective single-stranded DNA (ssDNA) proteins (SSB) and scaffolding factors (host IHF, Fis) (Figure 2).

The *origin* site plays a crucial role in the initiation of replication. *Origins* are cis-acting DNA sequences that serve as binding sites for initiator-dependent replisome assembly. Bacterial *origins* are determined by a single locus of specific initiator protein binding sequences, which are conserved through molecular evolution. Some of these sequences, as represented by DnaA boxes, display a highly conserved consensus motif that can be occupied by DnaA (the main bacterial initiator protein) throughout the cell cycle. The ATP-dependent self-assembly of several DnaA subunits occurs along the tandemly repeated DnaA box sequences located at the *origin* region. This reaction leads to the formation of an extended, higher-order helical oligomer with short local bends or large loops of the DNA. One of the final components to be loaded onto the replisome is the DNA polymerase holoenzyme (*E. coli* HolA, B, C, D, and DnaX in Figure 2), which is responsible for the synthesis of the new DNA strands, synthesis of primers (if a separate primer synthesis enzyme (primase) was not loaded earlier), and proofreading activity. There are also several crucial protein components (e.g., *E. coli* replication initiation regulatory proteins DiaA and Had, helicase loader DnaC, SeqA, Dam-methylase), which are required for the appropriate localization, stabilization, activation, and regulation of the entire replisome activity. In addition, there are DNA elements (DNA sequences, e.g., DUE) situated in close proximity or further away from the *ori* site to which regulatory and accessory factors bind [6]. Certain regulators, such as SeqA in *E. coli*, play a crucial role in sequestering *ori* (which provides a minimum space between forks generated during subsequent rounds of replication) through interaction with specific DNA sites that are localized with high density at the *ori* region. The recognition of these sequences depends on their methylation state, which is provided by another enzyme, DNA adenine methylase (Dam-methylase enzyme) [7,8]. SeqA protein associates with newly replicated DNA and forms complexes that follow the replication fork [7]. It has also been demonstrated that SeqA influences the *ori/ter* ratio through the regulation of the replication fork migration rate [9]. The Dam methylase mentioned above does not directly interact with the replisome. However, it influences the time-dependent activation of DNA replication by acting on DNA sequences at the *ori* region and thus affecting the binding of other proteins to these sequences. This way, the series of events involving several proteins’ release and binding of regulators result in topological changes along the *ori* region, enabling the DnaA initiator protein to bind and melt the *origin* DNA sequence [6]. This action, in turn, enables the loading of two replicative helicase molecules onto each of the single strands in the unwound region. As a result, two replication forks driven by a pair of replisomes, which move in opposite directions, are formed. This model of replication is known as the bidirectional DNA replication model (or the theta-like mode of DNA replication) [10]. In contrast, the unidirectional replication model involves a single replication fork driven by a single replisome, migrating in one direction. In addition to the above-mentioned DNA replication models observed in bacterial chromosomes, microorganisms exhibit less commonly found modes as well. The D-loop model, which is best described for mitochondrial DNA replication, as well as modes represented by linear chromosomes or those replicated via the rolling-circle route found in plasmid DNA [11], are examples of such models. The complexity of the *origin* region and initiation events is due to the chromosome replication and its coordination with the cell cycle, which is mainly controlled at the initiation step. Many of the protein regulators that are involved in the process of DNA replication have also additional roles in cell cycle regulation, ensuring a tight link between both processes. For example, *E. coli* DnaA, besides being the replication initiator, also serves as a transcription factor regulating the expression of genes that are involved in replication [12,13]. The involvement of a single agent, such as DnaA, in multiple cellular processes simultaneously significantly complicates the planning of an experiment when studying DNA replication, as well as the technical approach to the subject being analyzed [12,14,15,16,17].

After completing the initiation step, some of the regulatory or associated proteins dissociate from the DNA polymerase holoenzyme. The subsequent elongation phase of DNA replication is not subject to complex regulation, although there are many aspects of this phase that can be analyzed. During the elongation step, the DNA polymerase holoenzyme moves along the DNA template, while the RNA polymerase transcribes the required genes. Both processes, DNA replication and RNA transcription, occur simultaneously in the cell using the same template. It is well known that the transcription-mediated relaxation of chromatin plays a crucial role in the firing of replication *origins* in both prokaryotic and eukaryotic cells [18]. However, conflicts often arise between machineries carrying out these processes when DNA replication undergoes the elongation phase. Head-on encounters between the replication and transcription machineries on the lagging DNA strand can lead to replication fork arrest and genomic instability [19,20,21]. Replication forks can also stall, disintegrate, or collapse when they encounter DNA damage, DNA secondary structures, or tightly bound proteins [10,22]. Such a stalled fork can be subject to restart mechanisms [22,23,24,25,26]. Although a stalled replisome can last for a few minutes, it eventually falls apart [27]. According to Herrick and coauthors, coordinating the initiation and elongation processes is directly coupled with the regulation of the ribonucleotide reductase enzyme activity responsible for synthesizing necessary deoxyribonucleotides (in *E. coli* NrdAB) [28]. Although this enzyme is not directly connected to the replisome, its activity responds to stresses and changes in the metabolic state of the cell, tightly linking these processes. Due to various circumstances, the replication fork can disintegrate, resulting in double-strand breaks or double-strand ends, which can lead to chromosomal fragmentation. This situation requires restarting and repairing, which depends on homologous recombination mediated by the RecA recombinase, or a RecA-independent process involving RecBCD or other proposed models, such as replication fork reversal [29,30]. Given the circumstances under which the replication fork is reconstituted, the replication complex restarting replication may be another field of scientific research concerning DNA replication.

As mentioned above, the replication forks, which originate from the *oriC* region, travel in opposite directions along the chromosomal structure, and they do so at a speed of approximately 60 kilobases per minute to complete the replication round. The presence of a specialized replication fork trap, encompassing a sequence of polar blocks, restricts termination sites by preventing replisomes from leaving this region once they have entered it. The polar blocks develop due to the Tus terminator protein binding asymmetrically to a series of twenty-three base pair nonpalindromic *ter* sequences (*terA-J*) (Figure 3), distributed on both sides of the termination region. The Tus protein recognizes and binds asymmetrically to the *ter* sequences and forms a barrier for replication fork progression in an orientation-specific manner.

It is also important to consider the role of DNA topology in the correct progression of cellular processes, including DNA replication. Under standard cellular conditions, the negative supercoiling of DNA provides the energy for the localized, controlled melting of the DNA duplex. This allows for DNA and RNA polymerases, as well as other important enzymes, to access the internal nucleotide sequence of the double helix. Nevertheless, excessive negative supercoiling can result in the inhibition of bacterial growth and RNA degradation [31,32]. Consequently, alterations in DNA topology may serve as an indicator of blockage in DNA replication.

In addition, DNA synthesis in vitro and in vivo gives rise to intertwined, newly replicated DNA molecules. Catenanes, or links, are characterized by a right-handed (+), parallel structure. The incorrect unlinking of catenated intermediates of DNA replication results in problems with segregation, which in consequence leads to growth inhibition [32,33]. Thus, it is very important for a cell to control DNA supercoiling. The responsibility for the correct topological state of DNA belongs to cellular topoisomerases.

## 3. DNA Replication Process and Methodology for Its Investigation

Considering DNA replication, the most reliable methods for assessing such a crucial process in a cell are those that use the direct analysis of a living organism to capture (visualize) the situation at a given moment in the cell. There are only a few methods that analyze such a state inside the living cell, e.g., live-cell fluorescent microscopy. We term them in vivo methods. Unfortunately, the available magnification and resolution of the equipment at hand are the main limitations. This is the reason why currently available in vivo methods are unable to provide a complete and comprehensive understanding of what is occurring in living cells. Therefore, another approach is to fix the living cell in a way that closely reflects its state just before killing. A similar approach is to isolate material from a cell under conditions that keep it as intact as possible, again reflecting the state just before the cell was killed. We refer to these as in vivo post-mortem methods. Finally, there are a number of methods that can be used to analyze the products of in vitro reactions that simulate the situation that occurs in the living cell. These are referred to here as in vitro methods.

In taking into account the technical limitations of the equipment available, it is therefore necessary to use a combination of several techniques that allow an accurate assessment of the subject being evaluated. Some of the techniques require both, specialized equipment to be operated by qualified staff and specialists in data analysis. This concerns such techniques as electron or atomic force microscopy or mass spectrometry. Hence, we leave the nuances of such experiments to the specialists. However, there are several techniques that can be performed under standard laboratory conditions, despite their level of difficulty. Also, sometimes the use of specialized equipment requires several steps to be taken in a standard laboratory of molecular biology. It is important to note that in outlining the methodology presented in this paper, we have made the widely accepted assumption that “methods” are considered to be generally recommended ways of solving problems, while “techniques” refer to more specific ways of conducting research. Consequently, one research method often involves several different techniques, i.e., shorter single-purpose procedures.

It is also noteworthy that the employment of a single technique or even a single methodological approach allows for the acquisition of answers to queries pertaining to several stages or aspects of the replication process. Consequently, here at the beginning of each chapter presenting techniques or methods, a few numbered points are included. These points indicate which questions a given technique can answer, what data can be obtained with it, or which aspect of the replication process it analyses.

## 4. Experimental Design Stage—The Choice of Strategy and Primary Methods

The effective planning of the experimental strategy, that is, selecting the appropriate techniques and combining them into the methods that assure the realization of the research goal, is of crucial importance and should be thoroughly considered before starting one’s project. The core experiment can rarely be performed ad hoc; it usually requires several preparation steps to establish an experimental layout. Whether one decides to study DNA replication in vivo or in vitro, primary methods consist of chromosomal modifications (such as gene deletions, in-frame fusions, or point mutations) or the overproduction and purification of replication-involved molecules (proteins and DNA), respectively. In this section, we briefly describe the basic methodology of DNA manipulations leading to the construction of bacterial strains and DNA plasmid vectors.

### 4.1. Bacterial Strain Construction

Bacterial strains with a desired genetic background can be employed in the following ways:The generation of knockout mutants allows for the disruption or removal of genes, thereby facilitating the study of gene function and interactions between them [34].The CRISPR-Cas system enables the production of mutants with insertions or replacements. It allows the insertion of new genes or the replacement of native genes, facilitating the construction of strains with desirable traits, such as antibiotic resistance, metabolic pathway modifications, or the production of specific compounds [35].The construction of mutants bearing point mutations achieved with targeted single-base edits, which enable the investigation of specific amino acid changes on protein function or bacterial phenotype [36].Study of Regulatory Region Alterations. Alterations in promoters or other regulatory regions allow control over gene expression levels, which is valuable for understanding bacterial response to environmental conditions [37].

#### 4.1.1. Recombineering Methods of Genome Editing

Recombineering (recombination-mediated genetic engineering) is a precise and versatile method for constructing bacterial strains by introducing targeted modifications in their genomes. It employs homologous recombination to insert, delete, or replace specific DNA sequences using synthetic DNA fragments, thereby offering a highly precise and versatile approach to genetic engineering. A good tool for homologous recombination in bacteria is provided by bacteriophage lambda (λ), a virus that infects *E. coli* bacteria and is able to integrate its genetic material into the host genome by homologous recombination. Using its own λ-Red system, it is able to promote recombination between bacterial chromosomes and linear double-stranded DNA molecules with as little as ~50 (minimum 36 [38,39]) nucleotides of homology between the sequences in question. The genes of the system are not deleterious to the host cells and can therefore be induced in *E. coli* cells in isolation from other bacteriophage processes, either temporarily or permanently [40,41]. As a result, the λ-Red system is frequently employed to introduce modifications to the *E. coli* bacterial chromosome [39]. The λ-Red genes, typically under the control of an inducible promoter, can be introduced into the bacterial chromosome, for instance, in the form of a defective prophage [42], or delivered on a separate replicon. Additionally, a low-copy plasmid [38] is typically employed, which generally possesses a thermosensitive *origin* of replication. This enables the plasmid to remain inactive at higher temperatures, facilitating its removal from bacterial cells. Recombination signifies a transformative shift from traditional molecular cloning to precise, efficient genome engineering. It permits seamless genetic modifications, thereby enabling breakthrough advances in functional genomics, synthetic biology or biotechnology, and metabolic engineering. Currently, recombination remains a pivotal tool in bacterial genetics, undergoing constant evolution alongside complementary technologies such as CRISPR (Section 4.2), automation, and synthetic DNA synthesis.

#### 4.1.2. CRISPR-Cas-Based Genome Editing

CRISPR-Cas-based genome editing represents a powerful tool for the construction of bacterial strains, facilitating precise and targeted modifications to bacterial genomes. The CRISPR-Cas system, which was originally employed by bacteria as a defense mechanism, can be adapted for the introduction of specific genetic alterations. The process relies on the Cas9 protein, which acts as a molecular scissor, guided by a short RNA sequence (sgRNA) that is complementary to the target DNA sequence. Once introduced into a bacterial cell, the CRISPR-Cas system enables Cas9 to locate and cut the target DNA, thereby triggering repair mechanisms that permit insertions, deletions, or substitutions at the site. CRISPR-Cas-based editing is a widely utilized technique in the construction of bacterial strains [37,43]. While CRISPR-based editing in bacteria is highly precise, it can encounter difficulties. Some bacterial species may be less susceptible to CRISPR-Cas due to the absence of efficient transformation systems or robust repair mechanisms that neutralize CRISPR-induced cuts. Furthermore, although off-target effects are less prevalent in bacteria than in eukaryotes, they can still occur [44,45].

### 4.2. Vector Construction/Molecular Cloning-Restriction-Dependent and Independent Methods

Since the discovery of sequence-specific endonucleases named restriction enzymes (RE) in the early 1970s [46] and DNA ligase several years earlier [47], cutting and joining DNA fragments into recombinant molecules became possible. This process, termed molecular cloning, has revolutionized the life sciences [48]. Along with PCR prevalence, which allows us to amplify even small amounts of defined DNA fragments in vitro, molecular cloning is now considered a routine practice in laboratories worldwide. In principle, the restriction and ligation cloning method requires the cleavage of the insert and vector of choice with the same RE, joining them together with DNA ligase, followed by the transformation of bacterial cells, where the cloned vector is amplified. The proper choice of RE is one of the main guarantees of successful cloning. Commonly available plasmid vectors possess the so-called multiple cloning site, that is, the short DNA fragment containing plasmid-unique restriction sites for numerous REs [49]. Corresponding restriction sites for the insert sequence are usually added on the PCR primers. Importantly, the insert sequence must not contain internal restriction sites for the chosen RE. Although the commercially available RE pool consists of over 200 proteins, sometimes, it is still impossible to match the RE set for a given vector–insert pair. Moreover, after ligation, the restriction site is usually restored, resulting in a so-called ‘scar’ at the junction of the cloning fragments. It might be an issue, especially in case of translational fusions of a cloned sequence with other sequences present in a vector due to the higher risk of reading frame shift or the introduction of unwanted amino acid codons [50]. Seamless cloning is, however, possible within the restriction and ligation cloning when type IIS REs are used. These enzymes’ family cut DNA a few bases away from their recognition site (Figure 4A) [51].

Thus, generated overhangs vary in sequence and can be defined at the cloning design step, which gives an opportunity to seamlessly ligate even several DNA fragments in a one-pot reaction. This approach has been adapted by Golden Gate cloning, which gained popularity along with commercially available cloning kits (Figure 4B) [52]. Nevertheless, in all restriction-dependent cloning methods, the cloning site is limited by the presence of a particular RE site in the vector. To overcome this limitation, a wide variety of methods that do not follow traditional restriction–ligation procedures have been described. In principle, they can be divided into three groups—ligation-independent methods, recombination-based methods, and PCR-based methods (reviewed in detail in [50,53]). Obviously, each of them has its own advantages and constraints. The choice of cloning strategy is to some extent dependent on the laboratory experience and know-how of users, as well as the resources’ availability. Thus, we describe below a couple of cloning methods that we use on a daily basis and find particularly convenient in terms of both simple one-fragment insertions and more complex several-fragment assemblies.

Restriction-free (RF) cloning enables the insertion of any sequence at any location in the selected vector, without the need for its prior linearization and without introducing unwanted nucleotide ‘scars’ at the interface of the cloned fragments [54]. RF cloning is based on two consecutive PCRs. In the first reaction, the insert is amplified using primers with 5’-terminal overhangs to the vector sequence at the cloning site and then used as a so-called ‘megaprimer’ in the second reaction, in which the entire target construct is amplified (Figure 5).

The maternal plasmid, used as a template in the second PCR, is digested with the DpnI enzyme, and the reaction product is used to transform *E. coli* competent cells [54,55]. RF cloning is versatile and easy to design using free online tools [56]. It can be used for multicomponent assembly [57] and circular permutations [58]. However, when the target vector is very long (>10 kB), the second PCR is time-consuming and sometimes challenging. In such cases, Gibson assembly becomes a good alternative. Gibson assembly cloning is a single-step approach of joining several overlapping DNA fragments in a directed order and can be used with even a dozen or so fragments, which allows us to obtain large constructs [59]. This method relies on the activity of three different enzymes combined in a one-tube reaction. T5 exonuclease efficiently removes nucleotides from the 5’ end of DNA, leaving 3’ complementary overhangs that anneal together. The gaps between annealed fragments are then filled by Phusion DNA polymerase, followed by the ligation of resulting nicks with Taq DNA ligase [60]. Overlapping fragments for assembly are in most cases generated by PCR, though the Gibson reaction is performed at a constant temperature. The cloning vector needs to be linearized prior to reaction, usually by RE; nevertheless, the cloning position is defined solely by insert overhangs, not by the restriction site itself. Due to the fact that DNA polymerase needs to only fill up the gap sequences, the whole method lasts less than PCR-based cloning, especially for longer vectors. This makes Gibson assembly particularly useful in terms of generating large DNA molecules, such as synthetic (mini)chromosomes, that can be used as the substrate for DNA replication assays [61].

## 5. Molecular Biology Methods Used for Studying DNA Replication

To facilitate navigation in this article, methods collected in this section are presented in two groups (Figure 6): The first group contains techniques we call supporting—these are basic techniques that deliver a product required for subsequent experiments, or in some cases, they are part of more complex methods. The next group are the core methods that are divided according to the type of basic techniques: methods based on biochemical assays, methods based on image analysis, and methods based on electrophoresis (Table 1).

### 5.1. Supporting Techniques

Supporting techniques can be applied at various stages of experimental procedures. For PCR reactions, this approach is typically employed at the initial phase for constructing appropriate strains and finally for a preliminary verification of their structural accuracy. Protein purification enables in vitro experiments, and purified proteins are necessary for studies on protein–DNA and protein–protein interactions.

**The polymerase chain reaction (PCR) and DNA synthesis in vitro** are basic laboratory techniques used to rapidly produce millions to billions of copies of DNA for further detailed studies. Whether there is a need to amplify an insert for molecular cloning for the construction of a plasmid of interest or linear DNA that constitutes a target sequence for genome modification, PCR is indispensable. Now considered a routine technique and a golden standard in DNA amplification, it was first described only in 1985 by Kary Mullis [68] who was awarded a Nobel Prize for it in 1993. PCR relies on cyclic, rapid temperature changes that enable the successive denaturation of the DNA template, annealing of primers, and synthesis of the nascent DNA molecule by DNA polymerase-mediated primer extension. DNA polymerases used in PCR are thermostable enzymes that are active at elevated temperatures for a long time—this enables multiple rounds of exponential DNA amplification. The most widely known DNA polymerases used for PCR are the Taq polymerase and Pfu polymerase [69]. The first one possesses 5’-3’ polymerase activity and 5’-3’ exonuclease activity, but no 3’-5’ exonuclease activity, known as ‘proofreading’. Thus, the Taq polymerase cannot correct errors introduced during DNA synthesis, and therefore, its fidelity is lower than that of the polymerases’, with the proofreading activity like the Pfu polymerase. Moreover, Pfu polymerase generates blunt end products in contrast to Taq polymerase, which adds a 3′-adenine overhang to each end of the PCR product. Synthetic DNA fragments called PCR primers define a segment of the genome to be amplified. Around 15–18 3’-end nucleotides of primers have to be highly specific to the template sequence, whereas the 5’-end primer part can contain various additional sequences. This enables us to precisely adjust amplified DNA to one’s specific needs, i.e., adding specific overhangs to the genome- or plasmid-amplified DNA region or restriction enzymes to the recognition sites that enable the downstream processing of the amplified DNA [70].

**Identification and purification of DNA replication proteins.** The identification of proteins that form complexes with replication proteins can be achieved using specialized techniques such as mass spectrometry or Fourier transform infrared spectroscopy (FTIR). Mass spectrometry is a widely used technique for the identification of various proteins and their modifications. This technique focuses on molecular mass and fragmentation patterns, which allows for the identification of a given protein. Proteins are identified from the peaks in the acquired mass spectra using computational methods, where each peak theoretically corresponds to a peptide fragment ion [111,112]. Fourier transform infrared spectroscopy (FTIR) is an advanced spectroscopic technique that is employed to obtain absorption or emission spectra in the infrared range for solid, liquid, and gaseous samples. This technique gives information about the molecular bonds and functional groups present in a sample [113,114,115]. When there is no need to determine such advanced parameters, the purity and concentration of proteins employed by the methods described below can be assessed with the basic SDS-PAGE polyacrylamide electrophoresis technique [116,117]. The identified proteins can be then purified, as the availability of highly pure proteins is a crucial factor for the success of in vitro reactions. The isolation of specific proteins from a complex cell mixture is a process that usually takes advantage of the unique physical and chemical properties of proteins, including their size, charge, hydrophobicity, and binding affinity. This multi-step process, often time-consuming, aims to obtain non-denatured, homogeneous proteins in pure form and in sufficient quantities. For detailed protocols on the purification of proteins involved in DNA replication, readers are referred to the procedures described in references [118,119,120,121].

### 5.2. Core Experimental Methods

All the methods and techniques described below are summarized in Table 1.

#### 5.2.1. Biochemical Testing Methods

**Quantitative polymerase chain reaction (Q-PCR).** Q-PCR is a variant form of the classic PCR method by which the amount of the PCR product can be determined in real time. This in vitro method can answer the following scientific question:What is the *ori/ter* ratio in cells? Has a new round of replication already started?What is the DNA replication rate?Which genomic regions are being replicated during specific cell cycle phases?Has the amplification or deletion of genomic regions occurred?

By designing specific primers, many of the changes that arise during the replication process can be accurately detected and quantified using qPCR [71,122].

**Flow cytometry** is a powerful analytical technique widely employed in biology and medicine to assess the characteristics of individual cells or particles within a liquid suspension. In utilizing this method, it is possible to evaluate the following queries:Are there changes in cellular DNA content?How many cells are in a particular DNA replication phase?Is the DNA replication process disrupted?How do different cell populations respond to DNA replication inhibitors?

The technique utilizes a flow cytometer, a specialized instrument that passes cells or particles individually through a laser beam. As each cell or particle passes through the laser, it scatters light and emits fluorescence, generating data on various cellular properties, such as size, shape, granularity, and the presence of specific biomolecules, including proteins and DNA. Flow cytometry has emerged as a valuable technique in the investigation of eukaryotic cell attributes over the past few years. This method involves the labeling of cellular macromolecules, such as DNA, RNA, or proteins, using fluorescent dyes. Through this process, it becomes possible to precisely and expeditiously quantify the content of these macromolecules within individual cells [62]. Due to the small size of bacterial cells, the use of flow cytometry for their analysis was not initially feasible. It was not until 1983 when Boye and his team published research results acquired through the analysis of bacteria and demonstrated the potential of this technique. It has been shown that the fluorescence emitted by DNA stained with Mithramycin and/or Ethidium Bromide reliably indicates the DNA content in individual *Escherichia coli* cells [123]. Furthermore, the intensity of scattered light from individual bacteria is directly proportional to their cellular protein content and the effects of different antibiotics on bacterial growth and progression through the cell cycle can be evaluated efficiently and comprehensively. This surpasses the practicality of conventional methods such as PCR or Southern blotting [123]. Over the years, this method has been refined for working with bacteria. In 2012, Stokke and colleagues developed a Visual Basic-based simulation program for computing theoretical DNA distributions based on different choices of cell cycle parameters, such as the C and D phase durations and doubling time of bacterial cells [63,64,122].

**Assessment of DNA synthesis kinetics in vivo.** A method employing radioisotope-labeled nucleotides (^3^H-thymidine) provides answers to the following questions:How quickly does DNA synthesis occur in cells?What are the differences in DNA replication kinetics among different mutants?What factors influence the kinetics of DNA synthesis?

During the DNA replication process in vivo, bacteria absorb the labeled nucleotides from the medium, incorporating them into the DNA strand. The incorporation of the labeled nucleotide is quantified following cell lysis, employing a scintillation counter [65,66].

**Immunoassays.** In vitro techniques, such as Western blotting and enzyme-linked immunosorbent assays (ELISA) are classified as immunoassays, which can be used to answer the following questions:Which proteins are involved in the DNA replication process at its particular stages?How are the levels of replication-related proteins regulated during the cell cycle?How do various treatments or conditions affect the abundance of replication proteins?What are the interactions between replication factors?

These methods are widely employed for protein detection and quantification. They rely on the high specificity of antibodies to identify target proteins. Although immunoassays do not provide information about the amino acid sequence of proteins, they are effectively used to verify the presence of specific proteins in a sample. These assays are known for their sensitivity and accuracy, delivering rapid results. Moreover, they offer the advantage of not requiring advanced equipment or the use of radioactive labels, making them practical and accessible tools for various applications. The recognition of antigens by antibodies that have been conjugated to an appropriate marker is employed as a detection technique in more complex methods, for example, two-dimensional gel electrophoresis of DNA (described below) [67].

#### 5.2.2. Image Analysis Techniques

To investigate DNA replication in prokaryotes, imaging techniques can be used to visualize not only the DNA strands during replication but also individual proteins or protein complexes. These methods allow for a deeper understanding of replication dynamics by enabling the observation of molecular processes in real time. Among these, live-cell imaging has emerged as a crucial technique. Live-cell imaging allows for the visualization of biological processes within living cells, preserving their natural environment and enabling the study of dynamic events, such as the movement of replication forks, protein recruitment to DNA, and interactions between replication proteins [72,74]. In this review, we describe a range of imaging techniques, focusing on their application in the analysis of plasmid and chromosome replication, replication fork progression, *origin/terminus* dynamics, and the visualization of replication proteins.

**Wide-Field Epifluorescence Microscopy**—this technique involves illuminating the entire field of view with excitation light and collecting the emitted fluorescence. This microscopy technique enables the successful imaging of the following:Chromosome morphology;Replication fork progression in actively replicating regions;Replication protein localization and dynamics during DNA replication in vivo in live or fixed cells.

It is a user-friendly technique that does not require highly specialized equipment or skilled personnel. Samples can be easily prepared, and basic microscope skills are sufficient for operation [71,72,73]. On the other hand, confocal microscopy provides optical sectioning using a pinhole to eliminate out-of-focus light, improving image clarity. This technique is useful for visualizing cellular structures involved in DNA replication in three dimensions [74,75]. The preparation of samples follows a procedure similar to that of conventional fluorescence microscopy, while the operation of the microscope requires a higher level of training and expertise. Confocal microscopy analysis typically takes place in specifically equipped laboratories, under the guidance of specialized personnel.

**Electron microscopy (EM).** This is a powerful technique for studying biological material including the internal structure of cells at different molecular levels [124]. The technique enables us to perform the following:Evaluate the intracellular structure of bacteria (in vivo postmortem);Determine the DNA replication *ori* site;Assess the shape of the DNA replication intermediates;Assess the shape of DNA molecules;Identify a given protein binding site in the DNA molecule;Assess the protein complex shape.

Electron microscopes use an electron stream in a vacuum column, creating specific conditions for analyzed specimens. As a result, biological materials require complex preparation to survive these conditions and preserve the sample structures [76,125]. In the case of a transmission electron microscope (TEM), the image is produced by transmitting an electron beam through the specimen. The electron beam that emerges from the specimen carries information about its structure. As biological materials, such as nucleic acids and proteins, are almost electron-permeable, they require staining or coating with heavy metals like lead, uranium, or tungsten to scatter imaging electrons and create contrast between different structures. TEM has numerous applications in the study of DNA replication intermediates [33,77,78,126] and interactions between DNA and proteins [79,80,81]. Additionally, EM can provide supporting evidence for results obtained through other techniques, such as 2DAGE [82], footprinting assays [83,84,127], or EMSA [79].

##### Single-Molecule Imaging

The combination of single-molecule imaging with super-resolution microscopy has resulted in a paradigm shift in our capacity to study biological processes with an unparalleled degree of detail. There are several classes of single-molecule techniques, including force-based techniques and fluorescence-based techniques. These methods differ in their primary focus, the type of data they provide, and how they interact with the molecules under study.

Fluorescence-Based Techniques

These tools are a powerful asset for the study of DNA replication, offering comprehensive insights into the dynamic, real-time processes and molecular mechanisms involved in this process. These techniques allow for the following:Reconstruction of the DNA replication process in vitro (TIRF).Direct measurement of the speed at which DNA polymerases synthesize new DNA strands (TIRF, FRET).Observation of the manner in which replication forks may be paused or restarted, in response to such factors as DNA damage, secondary structures, and bound proteins (SMFM, TIRF, STED).Real-time visualization of sequential binding of initiator proteins, helicases, and other replication factors (TIRF).Determination of the *ori* sites (STED).Investigation of the binding and dissociation kinetics of replication proteins (SMFM, TIRF, STED) for detailed real-time kinetic analysis at the single molecule level, for high-resolution temporal studies of surface-immobilized systems with background noise reduction, and for the high-resolution spatial mapping and localization of protein–DNA interactions.

**Stimulated Emission Depletion Microscopy (STED)** achieves super-resolution by selectively deactivating fluorophores outside the focal spot, providing increased clarity in imaging [86,87,88,89]. **Single-Molecule Fluorescence Microscopy (SMFM)** is a technique that enables the visualization of individual fluorescently labeled molecules at the single-molecule level. SMFM allows researchers to observe individual molecules, providing insights into the behavior of DNA replication proteins and the dynamics of DNA replication at the single-molecule level in vivo, in live and fixed cells [74,88,89,128,129,130]. While STED and SMFM are distinct techniques, they can be combined to enhance single-molecule studies utilizing the super-resolution capabilities of STED to achieve the nanometer-scale localization of individual molecules. This integration offers unprecedented insights into DNA replication processes by providing high spatial resolution alongside the temporal dynamics captured by SMFM. SMFM experiments can be categorized into the passive observation and tracking of single molecules within a system or the active application of physical forces using optical or magnetic tweezers. The synergy between STED and SMFM holds significant potential for advancing our understanding of molecular mechanisms in complex biological systems [85]. Additional techniques, such as **total internal reflection fluorescence (TIRF)**, **Förster resonance energy transfer (FRET)**, and **fluorescence recovery after photobleaching (FRAP)**, can also be employed. TIRF and FRET are both microscopy techniques used to observe fluorescence phenomena. TIRF allows for the observation of fluorescence phenomena close to the sample surface. This technique can be used to simulate DNA replication in vitro. The procedure involves the coupling of the 5’ *terminus* of a DNA molecule (e.g., a rolling-circle substrate) to the surface of a specialized flow chamber. Replication components are then introduced into the flow cell (surrounding the DNA substrate). Simultaneously, a low concentration of intercalating stain is applied during the reaction. This enables the direct real-time imaging of the time-dependent length of growing DNA molecules [85]. The second technique, FRET, involves energy transfer between two fluorophores located in close proximity. This transfer is based on electromagnetic interactions between the fluorophores. TIRF is frequently used in SMFM to monitor the interaction of individual molecules with the sample surface [128]. FRET in SMFM can be used to measure the distance between two molecules, enabling the monitoring of structural changes within single molecules. This technique is especially valuable for examining protein interactions at the individual molecule level [129,130,131,132,133,134,135]. Usually, single-molecule tracking and super-resolution studies in bacteria rely on fluorescent protein fusion constructs. A number of fluorescent proteins are suitable for use in bacterial single-molecule super-resolution microscopy. However, in order to achieve successful results, it is necessary to pair them with specific activation control mechanisms. One example of such a mechanism is the use of photoactivatable fluorescent proteins, which can be readily converted from a non-fluorescent state to a fluorescent state by exposure to short-wavelength light pulses. This enables the user to exercise precise spatial and temporal control over the activation of fluorescence. Two well-known examples of photoactivatable fluorescent proteins are photoactivatable GFP and PAmCherry1 [87].

Force-Based Techniques

The measurement and application of forces to individual molecules represent pivotal approaches in understanding their mechanical properties. These techniques enable the precise quantification of molecular forces and facilitate the study of interactions and conformational changes at the single-molecule level. Among the various force-based methods, atomic force microscopy (AFM), magnetic tweezers, and optofluidic tweezers are particularly valuable for exploring the dynamics of biological molecules.

**Atomic force microscopy (AFM).** Atomic force microscopy (AFM) is a scanning probe microscopy technique used in nanotechnology and material science to image and characterize surfaces at the atomic and molecular levels. It was first developed in 1986 by Gerd Binnig, Calvin Quate, and Christoph Gerber. AFM is often used in conjunction with other techniques, such as scanning tunneling microscopy (STM). This non-optical technique employs the nanosensor of a sharp tip to scan the surface of a sample. In molecular biology studies, AFM can be used to analyze the topography of DNA, DNA–protein interactions, and the dynamics of DNA replication [136,137]; this technique can help to answer the following questions:What is the structure and organization of the bacterial replication fork?What is the topology of replication protein binding to DNA?What are the physical properties of bacterial DNA during replication?

There are several atomic force microscopy (AFM) types, each adapted for specific purposes and experimental conditions. Typically, the measurement does not require complex sample preparation procedures (compared to other microscopy methods) and can be performed in air, liquid, or vacuum environments. Among these, high-speed AFM (HS-AFM) has gained widespread use in the study of biomacromolecules, owing to its ability to capture dynamic processes in real time. HS-AFM facilitates the observation of molecular events, including conformational changes, enzymatic activities, and protein–DNA interactions, with both high spatial resolution and temporal precision, rendering it a particularly powerful tool for studying the mechanical and functional properties of biomacromolecules under near-physiological conditions [93,94,95].

**Magnetic tweezers.** This is a technique that enables the manipulation of single molecules, such as DNA or proteins, using a magnetic field. In experiments, a molecule is attached to a micrometric magnetic ball, whose position can be controlled by modifying the magnetic field. This allows for the molecule to be stretched or twisted, thereby studying its mechanical properties and interactions with other molecules. Magnetic tweezers are highly regarded for their straightforward design and capacity to manipulate multiple molecules simultaneously [96,97,98]. This technique can be useful for answering the following questions:What is the rate of DNA replication?Which proteins are involved in DNA replication, and how do they interact with each other?What are the mechanisms of DNA double-helix unwinding?

**Optofluidic tweezers.** This technique combines optical tweezers with microfluidic devices. In optical tweezers, a laser is used to trap and manipulate small particles in three dimensions, while microfluidic devices control fluid flow on a microscale. This combination creates an advanced tool for precise manipulation and analysis at the level of individual cells or molecules. Known as “optofluidic tweezers”, this integration enhances the efficiency and versatility of manipulations in biological and biophysical research [99,100,101,102,103]. Using this method, researchers can explore the following questions:How do external factors influence replication dynamics?What are the dynamics of replication in complex systemsHow does the replisome function under different conditions?

In addition to atomic force microscopy (AFM), magnetic tweezers, and optofluidic tweezers, there are several other force-based techniques that further expand the toolbox for studying molecular mechanics. These include optical tweezers, micropipette aspiration, and magnetic resonance force microscopy (MRFM), which combine principles of AFM and magnetic resonance to achieve nanoscale force measurements with high sensitivity. These advanced techniques, in conjunction with those previously discussed, underscore the diversity of tools at researchers’ disposal for exploring molecular forces and provide complementary perspectives on the mechanical and functional properties of biological systems.

#### 5.2.3. Electrophoretic Techniques

The agarose and polyacrylamide electrophoresis of DNA, RNA, and protein are the major basic techniques used for research at the molecular level. Most methodological approaches to the subject of analysis involve the use of the classical type of electrophoresis for the identification or purification of DNA fragments. There are many publications describing these technics in detail [106,138,139]. Therefore, here, we will only focus on a few intricate types of electrophoretic techniques used in studies of DNA replication.

Manipulating electrophoretic parameters enables the acquisition of diverse data on DNA structure. The outcome of the experiment depends on various factors, such as DNA intercalators, denaturing conditions, gel concentration, applied voltage, run time, and the orientation of the electrophoresis run. The choice of parameters determines the type of electrophoresis and the expected results (Figure 7).

The DNA intercalating factors are some of the parameters that can influence the nature of the electrophoresis result. DNA intercalative agents such as chloroquine or ethidium bromide locally unwind the double helix. These agents introduce positive super-helical twists to covalently closed supercoiled molecules, resulting in topological changes in the DNA molecules, which cause them to migrate differently in a gel. The agarose electrophoresis of DNA (performed in one or two dimensions) in the presence of chloroquine in the gel and electrophoretic buffer allows us to study changes in the DNA topology and analyze the distribution of the linking number in DNA molecules [140]. Consequently, alterations in DNA topology may serve as an indicator of blockages in DNA replication.

**Two-dimensional agarose gel electrophoresis of DNA** (2DAGE or 2D electrophoresis) is a method dedicated to analyzing the intermediate structures of DNA formed during replication at each stage. Using this method, we can obtain answers to the following questions:At which location within the analyzed molecule does the formation of the replication bubble indicate the position of the *origin* of replication?At what location does the termination of the replication round occur in the molecule?In which direction does the replication fork migrate?Is there any blockage of replication fork migration? If so, at which point within the molecule?According to which model does the DNA molecule replicate?Is there any recombination between the molecules analyzed?What is the topological state of the DNA molecule?

It is assumed that there are three main modes of DNA replication (theta, rolling-circle, and D-loop). Each of these modes is characterized by specific shapes of the DNA molecule undergoing replication (so-called replication intermediates). Undertaking the 2DAGE electrophoretic method may be very ambitious since the DNA intermediate structures are usually unstable. They are characterized by the presence of single-stranded DNA (ssDNA), which, when extracted from the cell and purified of the proteins that stabilize them within the cell, are highly susceptible to breakage. Therefore, the preparation of DNA samples is a very important step, and thus, appropriate protocols for isolation have been published [104,105]. These procedures enable the isolation of all DNA, chromosomal or plasmids, irrespective of whether the molecule is undergoing replication or not. Therefore, it is important to ensure optimal growth conditions to maximize the number of replicating molecules at the time of isolation, i.e., using synchronized cell cultures. 2DAGE can be applied to assess topological changes in uncut forms of DNA [104,105] but also to the restriction fragments of the DNA. The standard 2DAGE protocols are set for linear DNA about 3–6 kb in size [141]. There are several modifications for larger fragments and also for application to human DNA. The majority of modifications concern the agarose gel concentration, the duration of electrophoresis, and the strength of the applied voltage [142]. The planning digestion of analyzed DNA with restriction enzymes is crucial to obtain relevant outcome images of the molecule undergoing replication process. The optimal scenario would be to designate a hypothetical position within the DNA molecule, such as *ori*, *ter*, or a potential impediment to replication fork migration. Once the location of the sequence in the DNA fragment undergoing electrophoresis is known, specific images can be expected that will either confirm or disconfirm the hypothesis. In the case that the location of a specific sequence cannot be proposed, an effective approach is to select restriction enzymes that will allow for the analysis of different yet overlapping regions of the DNA molecule in the various samples subjected to electrophoresis.

Important information concerning chromosome dynamics and processes to which DNA is subjected can be obtained depending on the conditions set during the electrophoretic run. This results in a variety of 2DAGE variants, which are outlined below.

**Neutral**–**neutral 2D agarose gel electrophoresis** (when DNA samples are run in neutral, TBE, or TAE buffers) performed under standard conditions. Depending on whether the examined DNA molecule is in native or linearized form, the 2DAGE method can provide information on the following:The position of *ori* sites [143];The position of *ter* sites [144,145];The type of DNA intermediates created during replication;The positions of potential barriers, pauses, or stops in the replication fork movement [146,147];The topological state of the plasmid DNA [148];The torsional tension of replication intermediates [149].

If DNA molecules isolated from a cell, which are at different stages of replication, are linearized at known sites, then the branched DNA fragments specific for DNA replication models can be obtained (e.g., so-called replication bubbles, Y-shaped or X-shaped DNA fragments). The electrophoresis described herein is conducted under conditions that facilitate the separation of linearized DNA replication intermediates in a characteristic, well-described manner [150,151,152]. In the first electrophoresis (the first dimension), the DNA restriction fragments are separated according to the molecular mass of the DNA fragments. This separation mode is achieved using low voltage, long run times, and a low concentration of the gel. In the second electrophoresis (the second dimension run perpendicular to the first dimension), high voltage and a high gel concentration are applied, and ethidium bromide is added to the gel and electrophoretic buffer. The use of such conditions allows for the separation of DNA according to the shape of the DNA molecules. The result is that each type of branched DNA fragment is positioned within the gel in the form of a distinctive arc. A comprehensive analysis of the data provided by the 2DAGE of different restriction fragments of the molecule can lead to the designation of the DNA replication model and may indicate even minor changes in DNA replication intermediates resulting from genetic or metabolic deviations from the basic cell state. The separation of molecules of the same mass but different shape, as used in this method, also enables a more detailed analysis of the topology of DNA plasmids than that obtained by electrophoresis in one direction. In this case, non-linearized DNA molecules are subject to analysis and the addition of an intercalating agent (e.g., chloroquine) during the separation process, which enables a more effective separation of populations of different topoisomers [148,149].

**Alkaline 2D electrophoresis (2D electrophoresis in denaturing conditions**)—this type of technique allows for the detection of single-strand breaks (SSBs) on dsDNAs. This approach has been used for studies of DNA replication and recombination [150,153,154]. In the first dimension, dsDNAs are separated in a neutral buffer, such as TAE and TBE. The second dimension of electrophoresis is performed under alkaline conditions, which cause the denaturation of dsDNA and separation of ssDNAs. The structure of the replicating molecules affects the shape of the nascent strand arcs arisen on a gel after electrophoresis.

**Pulsed-Field Gel Electrophoresis (PFGE)**—this type of electrophoresis is dedicated to huge bacterial chromosomes. The qualitative and sensitive nature of this method allows for the following:Detection of double-strand breakage that occurred in chromosomal DNA;Observation of DNA dynamics during DNA replication when samples drawn over time are analyzed.

The gel matrix has limited molecular sieving effects, which prevent the separation of DNA molecules larger than 50 kb. To overcome this limitation, pulsed-field gel electrophoresis (PFGE) was developed. There are several protocols concerning pulse-field electrophoresis development and improvement [106,155,156,157,158]. PFGE enables the separation of large DNA molecules in an agarose gel by periodically altering the direction of the electric field relative to the gel. This technique can be applied to both circular chromosomes and linearized large DNA molecules (up to 12 megabases) [106]. Contour-clamped homogeneous electric field (CHEF) gel electrophoresis is the most frequently used system for PFGE analyses. This system applies a contour-clamped homogeneous electric field that alternates between two orientations with a single pulse time [109]. The pulse time primarily determines the size range of separation—longer pulse times enable the separation of larger DNA fragments. The migration rate of large linear DNA molecules resolved during PFGE can be affected by various conditions, such as switch times, electric field strength, time of electrophoresis, buffer temperature, and agarose concentration. Circular DNA does not penetrate the gel due to its large size and conformation. Hence, PFGE can be used to separate chromosomes with double-strand breakage, while circular bacterial chromosomes need to be digested by restriction endonucleases before separation by PFGE. Large chromosomes are highly susceptible to breakage, similarly to replication intermediates. Therefore, it is crucial to safeguard the DNA from physical shearing forces and endo- and exonucleolytic cellular agents during isolation. Appropriate procedures are available [106].


**Electrophoretic techniques concerning identifying DNA-binding protein interactions.**


**E**lectrophoretic **M**obility **S**hift **A**ssay **(EMSA)** (or the band-shift assay) provides answers to the following questions:Does the agent bind to the DNA fragment being analyzed?Does the protein bind to the DNA fragment alone or in a complex with other proteins?What are the optimal binding conditions for the DNA–protein interaction?

EMSA involves the binding of a protein to a nucleic acid fragment, causing a retardation in its electrophoretic (agarose or polyacrylamide gel) run compared to the same nucleic acid not bound by a protein [109,110,159]. This assay required first purifying the protein of interest. The analyzed DNA fragments are usually labeled with isotopes, covalent or noncovalent fluorophores, biotin, or visualized under UV light after a prior use of intercalating dyes. These components are then used in a DNA–protein binding reaction, and the resulting products are separated using non-denaturing gel electrophoresis. Labeled DNA can be then detected by autoradiography, fluorescence imaging, chemiluminescent imaging, and/or chromophore deposition.

**DNA footprinting assay**—This technique allows for the following:Assessment of sequence selectivity of DNA-binding proteins or other ligands [107,108].Assessment of the order in which several proteins making a large complex bind to the DNA.Assessment of competition between proteins binding to the same DNA region.

This method is based on the ability of the binding ligand to protect double-stranded DNA from cleavage at its binding site. A chemical or enzymatic cleavage agent is used to cut a double-stranded DNA fragment; most commonly, DNase I or hydroxy radicals are used. This results in a random distribution of products that can be resolved on a denaturing polyacrylamide gel. The monitoring of the separated fragments is possible because the DNA fragment is labeled at one end of one strand (usually the ^32^P isotope, but also covalent or noncovalent fluorophores or biotin are used) [109]. The cleavage agent does not have any sequence selectivity, ensuring single digestion within a DNA fragment. Another reaction mixture contains the same double-stranded DNA fragment, but before its digestion with the cleavage agent, a sequence-selective ligand/protein is added. If the ligand has an affinity for the DNA fragment, it protects the bound regions from cleavage. As a result, a gap or ‘footprint’ in the pattern of the cleaved DNA is observed on a gel. To predict the precise location of the ligand binding site, a DNA sequencing reaction can be run alongside the samples on the polyacrylamide gel. Sanger DNA sequencing is often used in this case. The footprinting technique can also be used to examine the kinetics of ligand–DNA interactions by establishing different DNA–ligand incubation times [107]. On the other hand, the comparison of band intensities within the footprint at varying ligand concentrations enables quantitative footprinting analysis. This method also provides an estimate of the ligand dissociation constant at each binding site [107].

## 6. Conclusions

We realize that as technology progresses, increasingly sophisticated new molecular biology techniques are being devised. While such novel molecular approaches have the potential to enhance data acquisition, established techniques that are well defined and understood offer a reliable and cost-effective way for obtaining information. An example of modern advanced technology that should be mentioned here is systems biology. It offers a comprehensive examination of the molecules that constitute cells. Genomics provides information on the entire genome, transcriptomics offers insights into the timing and the transcription activation of genes in a cell, proteomics concerns the analysis of the full set of proteins present in a biological sample, and metabolomics encompasses the examination of endogenous metabolites within a cell. All these technologies require specialized equipment, software, and bioinformatic competence, which is typically provided by commercial entities. Each of these areas can provide a wealth of valuable information, and when considered together with the methods described in this article, these will allow for a broader understanding of the problem being analyzed.

## Figures and Tables

**Figure 1 ijms-26-00446-f001:**
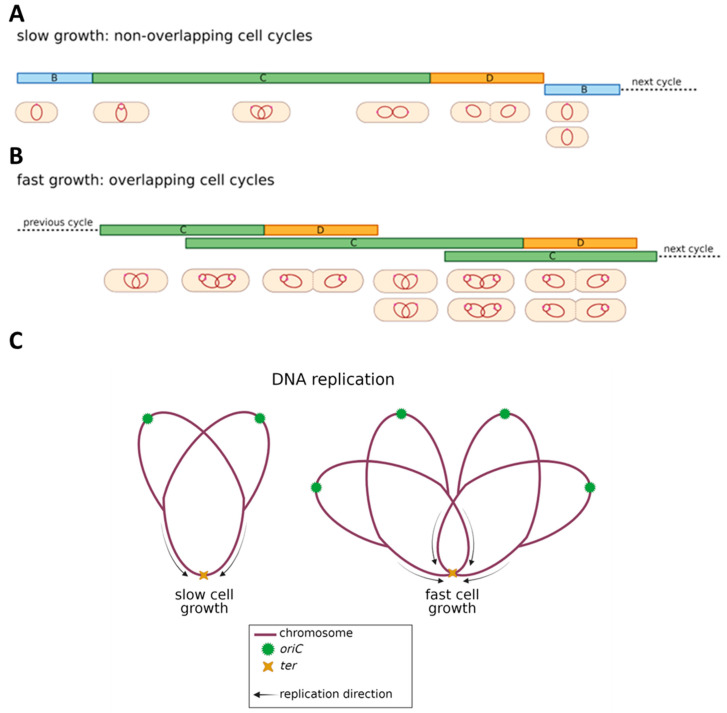
Cell cycle and DNA replication: slow (**A**) and fast-growing (**B**) bacterial cells. Panel (**C**) depicts an *origin*-to-*terminus* ratio in slow and fast-growing bacteria. B phase-blue colour, C phase green colour, D phase-orange colour.

**Figure 2 ijms-26-00446-f002:**
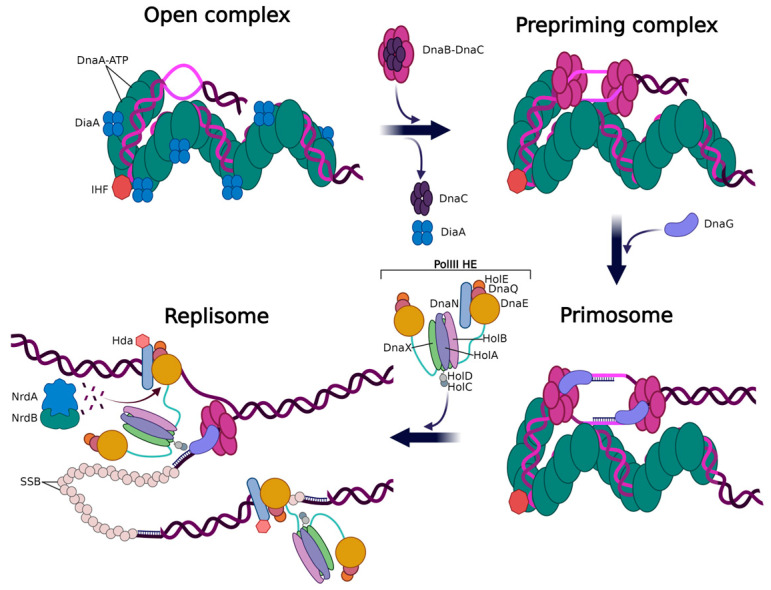
Steps in the assembly of the replication complex in *Escherichia coli*, covering events from the recognition of the *ori* sequence by the initiator protein DnaA, DnaB helicase assembly, the local unzipping of the double-stranded DNA structure (light purple DNA strands) to the primer synthesis by the DnaG primase, and the early elongation of both strands on the DNA template. PolIII HE- —DNA polymerase III holoenzyme; NrdA, NrdB-subunits of ribonucleoside diphosphate reductase 1; DiaA, had -replication initiation regulatory proteins. DiaA. stimulates DnaA assembly on *oriC*, whereas Hda inhibits the reinitiation of DNA replication by promoting DnaA-ATP hydrolysis in the process termed RIDA. Thick, black-shaded arrows indicate the transition to the subsequent stage in the initiation of DNA replication. Thin black arrows indicate the attachment or detachment of additional proteins/complexes during the subsequent stages in the formation of the replication complex. See text for more details.

**Figure 3 ijms-26-00446-f003:**
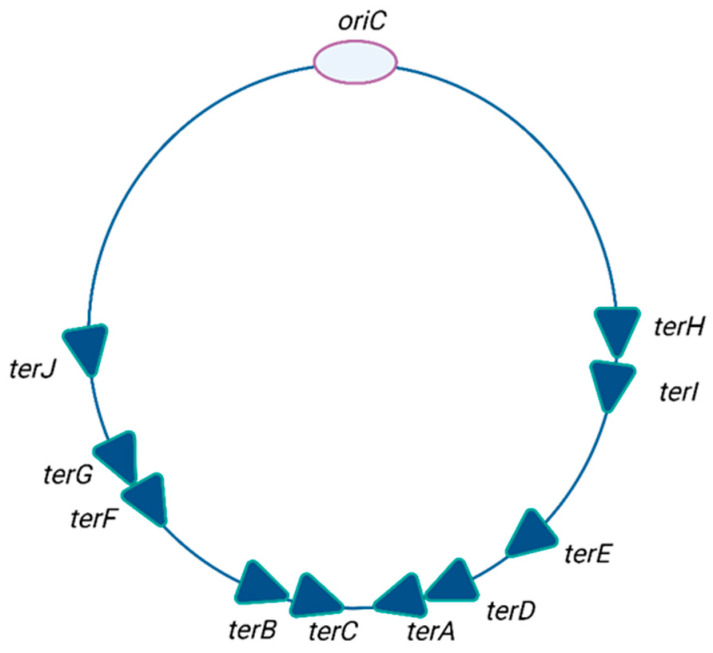
Distribution of *ter* sites in the *Escherichia coli* chromosome. Light purple indicates single-stranded DNA fragments in locally separated (unwound) DNA double helix at the *ori* site. The dark turquoise line of the circle marks the DNA helix of the chromosome.

**Figure 4 ijms-26-00446-f004:**
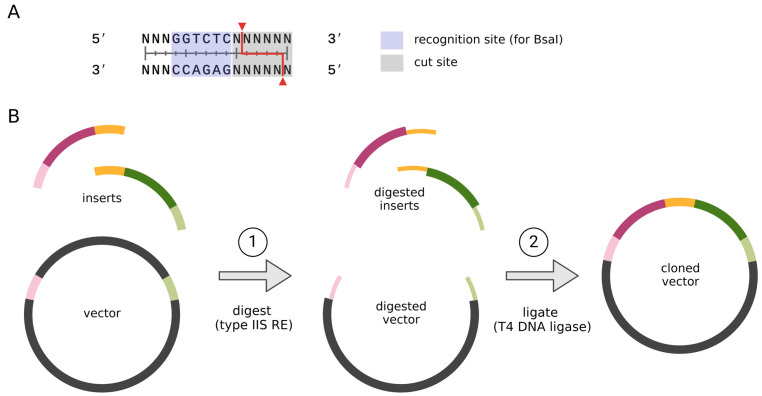
(**A**) Type IIS RE cutting scheme. (**B**) The Golden Gate cloning scheme. The successive stages of the experiment are indicated by numbers. First, vectors and insert(s) are digested with type IIS RE of choice, which gives the complementary sequence overhangs (step 1 indicated with gray arrow) that can be ligated using T4 DNA ligase to a cloned vector (step 2 indicated with gray arrow). Since type IIS RE is cut outside the recognition sequence, the specific overhangs can be freely designed, which gives the possibility to assemble several fragments in a one-pot reaction. DNA fragments of the vector and insert complementary to each other were marked with identical colours. Ligation occurs between fragments of the same colours. The shorter fragment of the vector DNA located between the complementary sequences highlighted in colour, is cut out during restriction digestion.

**Figure 5 ijms-26-00446-f005:**
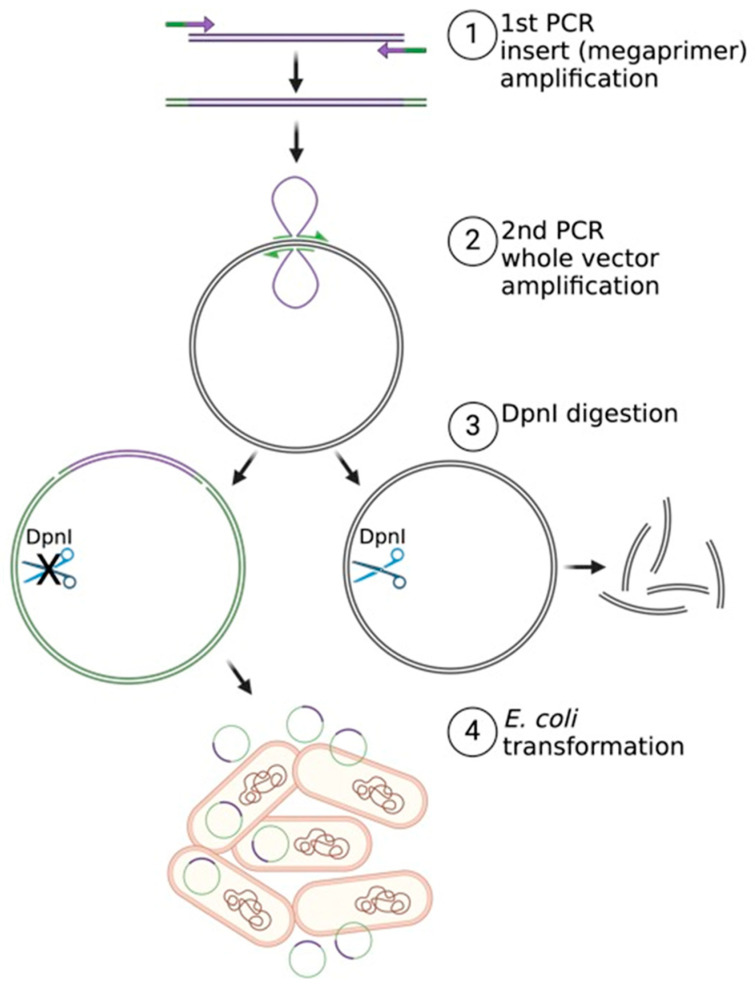
Restriction-free (RF) cloning scheme. The successive stages of the experiment are indicated by numbers. The first two stages, numbered 1 and 2, represent two consecutive PCR reactions. The third stage, numbered 3, involves the selection of the product by digestion with the DpnI enzyme. The fourth stage, numbered 4, entails the transformation of *Escherichia coli* cells with the obtained construct. Overhang-containing primers are indicated by thin arrows. Thick arrows indicate the transition between subsequent cloning steps, describes as numbers above.

**Figure 6 ijms-26-00446-f006:**
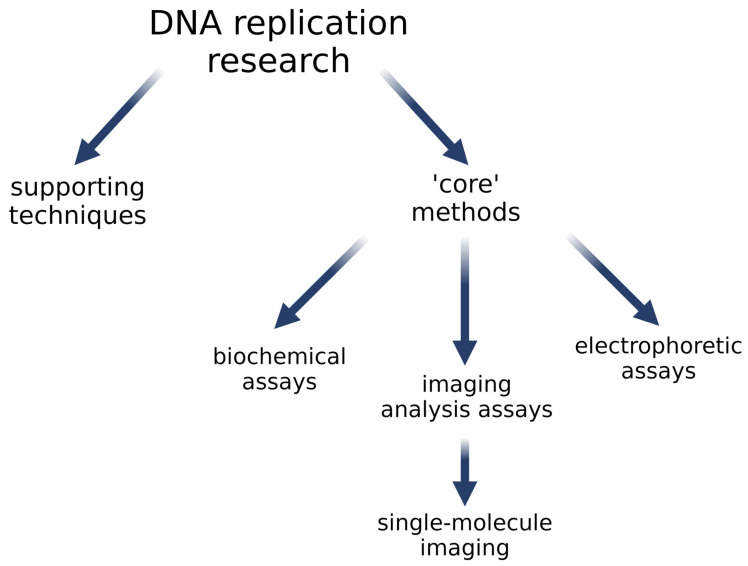
Groups of methods taken into consideration in this paper—see details in the text.

**Figure 7 ijms-26-00446-f007:**
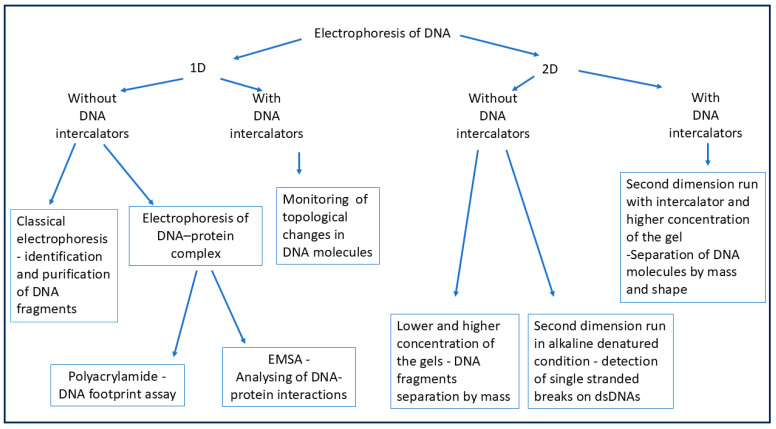
Electrophoretic-based methods considered in this paper. A brief characterization of each method is included. See text for details.

**Table 1 ijms-26-00446-t001:** Core techniques/methods employed in the study of DNA replication. Please see the text for further details.

Biochemical Techniques							
Method	Application	Target Material	DNA Replication Stage	Impediments/ Contraindication	Advantages	Level of Assessment	References
Flow cytometry	- Determination of major cell characteristics (size, shape, granularity) - Analyses of relationship between the cell cycle and DNA replication - Assessment of the current state of DNA replication and cell cycle and their progression	- A single living bacterial cell with fluorescently labeled DNA, RNA, or other cellular components	- Each stage of DNA replication and of the cell cycle	- Specialized equipment - Small cells to gate - Long time for RNA digestion required - Very good quality equipment required to observe small cells (flow cytometer)	- Easily prepared samples - Rapid and high-throughput analysis - Multiparametric capability - Versatility in fluorescent probes - Sorting capability (FACS)	in vivo postmortem	[62,63,64]
Kinetics of DNA synthesis in vivo	- Assessment of DNA synthesis during bacterial culture growth.	- Bacterial culture	- Each stage of DNA replication and of the cell cycle	- Specialized equipment required - Radiolabeled precursors	- Easily prepared samples - High sensitivity and specificity - Easy data analysis	in vivo	[65,66]
Immunoassays	- Identification of proteins, protein-DNA complexes, and specific forms of nucleic acids in situ		- Each stage of DNA replication and of the cell cycle	- Availability of antibodies against identified molecules	- High specificity for target molecules - Sensitive detection of low-abundance proteins - Multiplexing capability	in vivo post mortem	[67]
Quantitative real-time PCR and DNA synthesis in vitro	- DNA replication rate by *ori/ter ratio* analysis - Initiation rate		- Each stage of DNA replication and of the cell cycle		- High sensitivity and specificity - Quantitative analysis - Multiplexing capability - Rapid results - Minimal sample requirements - Easy data analysis	in vitro	[68,69,70]
**Image analysis techniques**							
Method	Application	Target Material	DNA Replication Stage	Impediments/ Contraindication	Advantages	Level of Assessment	References
Wide-Field Epifluorescence Microscopy	- Determination of the major cell characteristics (size, shape, granularity) - Visualization of cellular structures - Visualization of individual fluorescently labeled molecules	- A single living bacterial cell with fluorescently labeled DNA, RNA, or other cellular components	- Each stage of DNA replication and of the cell cycle	- Specialized equipment required	- High temporal resolution - Versatility in fluorescent probes - Colocalization studies - Enhanced spatial resolution	in vivo, in vitro	[71,72,73,74,75]
Electron microscopy	- Identification of the shape of replicated DNA and recombination intermediates, - Visualization of DNA-protein complexes in the in vitro reaction	- DNA isolated from cells - In vitro reaction mixture of DNA and analyzed proteins	- Initiation - Elongation (from early to terminal stages)	- Fragility of DNA replication intermediates, leading to loss of material or misinterpreted results - Specialized equipment with trained staff required - Complex preparation process with additional equipment required	- Analyses of the single molecule	in vitro, in vivo postmortem	[76,77,78,79,80,81,82,83,84]
Stimulated Emission Depletion Microscopy (STED)	- Visualization of replication *origins* and *termini* - Tracking replisome dynamics - Protein–DNA interactions in the nucleoid - Spatial organization of chromosomes during replication	- A single living bacterial cell with fluorescently labeled DNA, RNA, or other cellular components	- Each stage of DNA replication and of the cell cycle	- Specialized equipment required	- High temporal resolution - Versatility in fluorescent probes - Colocalization studies - Enhanced spatial resolution - Visualization of biological processes in real time	in vivo in vitro	[85,86,87]
Single-Molecule Fluorescence Microscopy (SMFM)	- Visualization of individual fluorescently labeled molecules	- A single living bacterial cell with and without fluorescently labeled DNA, RNA, or other cellular components	- Each stage of DNA replication and of the cell cycle	- Specialized equipment required	- Single-molecule resolution - Real-time dynamics - Minimal sample preparation - Compatibility with super-resolution techniques	in vivo in vitro	[85,86,87,88,89]
TIRF, FRET, FRAP, Single-Molecule Imaging	- Study of molecular and cellular phenomena at any liquid–solid interface - Real-time visualization of in vitro reconstituted replication of individual DNA molecule - Determination of the activity and dynamics of individual polymerases during coordinated leading- and lagging-strand synthesis - Examination of the replisome stoichiometry and architecture in living cells	- Isolated DNA, RNA, or other cellular components - Bacterial cell	- Each stage of DNA replication	- Specialized equipment required	- High-contrast imaging - of single molecules - Real-time dynamics - Minimal photodamage - Spatial precision - Compatibility with various fluorophores	in vivo in vitro	[85,90]
Atomic Force Microscopy (AFM)	- Imaging and characterization of protein or DNA surfaces at the atomic and molecular levels	- DNA or protein	- Each stage of DNA replication	- Specialized equipment required - Good-quality samples required	- High-resolution imaging - Direct visualization of DNA and proteins - Real-time dynamic observations - No requirement for labeling or staining - Versatility in sample preparation	in vitro, in vivo postmortem	[91,92,93,94,95]
Magnetic tweezers	- Dynamics of replisome assembly and function - Effects of DNA topology on replication - Protein–DNA interactions	- DNA or protein	- Each stage of DNA replication	- Specialized equipment required - Good-quality samples required	- Non-invasive and precise control - Long-term observation—perfect for studying slow processes due to the high stability of the setup	in vitro, in vivo postmortem	[96,97,98]
Optofluidic tweezers	- Protein–DNA interactions	- DNA or protein	- Each stage of DNA replication	- Specialized equipment required - Good-quality samples required	- High precision and versatility - Non-invasive manipulation - Integration with other techniques such as super-resolution microscopy or fluorescence imaging	in vitro, in vivo postmortem	[99,100,101,102,103]
Electrophoretical techniques							
Method	Application	Target material	DNA replication stage	Impediments/contraindication	Advantages	Level of assessment	References
2Dgel	- Identification of the shape of replicated DNA and recombination intermediates - Determination of DNA replication modes - Determination of DNA replication initiation sites - Assessment of alterations in the DNA topology	- DNA isolated from cells - DNA in vitro reaction	- Elongation (from early to terminal stages)	- Fragility of DNA replication intermediates, leading to loss of material or misinterpreted results - Multi-stage procedure	- The possibility of detecting different shapes of DNA molecules even with slight differences in length and shape	in vivo postmortem, in vitro	[104,105]
Puls-Field Electrophoresis	- Identification of the shape of replicated DNA or recombination intermediates in case of large genome molecules	- DNA isolated from cells	- Each stage of DNA replication	- Requires special apparatus	- Detection of large DNA molecules, e.g., chromosomes	in vitro	[106]
DNA footprinting assay	- Assessing the sequence selectivity of DNA-binding proteins or other ligands	- DNA and protein/ligands from in vitro reaction	- Each stage of DNA replication	- Requires prior purification of the protein of interest	- Precise determination of the DNA sequence to which the protein binds	in vitro	[107,108]
EMSA	- Assessment of the protein’s affinity for binding a specific DNA fragment	- DNA and protein/ligands from in vitro binding reaction	- Each stage of DNA replication	- Requires prior purification of the protein of interest	- No special apparatus needed	in vitro	[109,110]

## Data Availability

Not applicable.

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
