# Peer review of "Concise Overview of Methodologies Employed in the Study of Bacterial DNA Replication"

_ijms, 2025, doi:10.3390/ijms26020446_

Round 1
Reviewer 1 Report
Comments and Suggestions for Authors
The review by MaciÄ…g-DorszyÅ„ska et. al. provides an introduction to prokaryotic DNA replication, with a focus on the methodologies and techniques used to study the process. The authors believe that readers will benefit through selecting an appropriate research methodology. In Section 1 and 2, the authors introduces steps in the assembly of the replication complex in Escherichia coli, starting from recognition of ori sequence by the initiator protein DnaA, followed by DnaB helicase assembly, unwinding of the double-stranded DNA structure to the primer synthesis by DnaG, and finally elongation of both strands of the template DNA by PolII holoenzyme. Section 3 covers the methodology/principle of bacterial strains construction- like recombineering methods, using CRISPR, and molecular cloning methods– both restriction enzyme-dependent and independent. Section 4 talks about the various core and supporting techniques to study the process of DNA replication, involving techniques like, but not limited to, kinetics of DNA synthesis in vivo, quantitative real-time PCR and DNA synthesis in vitro, various microscopic techniques, DNA footprinitng assays, EMSA, etc.
Overall, this review covers broadly the field of various methodologies in bacterial DNA replication. Given the fact that authors want this article to be a road map, they should elaborate on all the techniques and provide more details on the PCR methods. The authors have written the electrophoretic section very well and with intricate details. However, the authors could provide a schematic of PCR techniques they mention, like Golden gate. These elaborations could involve the addition of schematics that can be in the Supplementary Material. In addition, it is essential that they include a detailed section on single-molecule techniques, their advantages and limitations, and, add this to the techniques Table. These techniques include magnetic tweezers, optical tweezers combined with microfluidics and finally the rolling circle DNA replication assay combined with TIRFM.
Minor points:
Table 1 can have an additional column for references of all the claimed statements, presumably the seminal papers from which the claims, outcomes and limitations are taken.
AFM – question 2 can be rewritten as Topology of binding of replication proteins, since AFM is a surface tapping technique. Also, it would be helpful to the reader if the authors specifically mention the High Speed – AFM as it has a widespread use in biomacromolecule regime.
Please correct the following typographical errors
Line 299 – “which is was”
Line 326 – “in in”
Line 328 – “revolutionized the life sciences”
Line 374 – “Transformed into Ecoli cells”
Line 475 – “the analysis of bacterial demonstrated”
Line 621 – “technics”
Line 702 – “ethidium bromide are added”
Author Response
Thank you very much for all your comments and corrections. We have taken care to include them all in the manuscript. All changes to the text of the manuscript have been put in blue font.
The detailed responses to the reviewers' comments are provided point by point below:
“...they should elaborate on all the techniques and provide more details on the PCR methods.”
We have provided more details in the chapters on PCR and single-molecule methods. The diagram concerning Golden gate cloning has been added (Fig 4). This operation has resulted in the renumbering of the figures.
“These elaborations could involve the addition of schematics that can be in the Supplementary Material. “
Since review papers provide no supplementary appendix, additional diagrams are incorporated into the main body of the text.
“In addition, it is essential that they include a detailed section on single-molecule techniques, their advantages and limitations, and, add this to the techniques Table. These techniques include magnetic tweezers, optical tweezers combined with microfluidics and finally the rolling circle DNA replication assay combined with TIRFM.”
As suggested, we have created a new subsection on the aforementioned content. This change resulted in the renumbering of the chapters. Also relevant to the text, we have added these techniques in Table 1.
“Table 1 can have an additional column for references of all the claimed statements, presumably the seminal papers from which the claims, outcomes and limitations are taken."
The column “References” is created in Table 1.
“AFM – question 2 can be rewritten as Topology of binding of replication proteins, since AFM is a surface tapping technique."
Question 2 has been changed as suggested.
“Also, it would be helpful to the reader if the authors specifically mention the High Speed – AFM as it has a widespread use in biomacromolecule regime.”
As suggested, we have mentioned the High Speed-AFM
“Please correct the following typographical errors”
Thank you very much for pointing out the errors in the text. We have carefully made the corrections.
Reviewer 2 Report
Comments and Suggestions for Authors
see attached

see attached
Author Response
Thank you very much for all your comments and corrections. We have taken care to include them all in the manuscript. All changes to the text of the manuscript have been put in blue font.
The detailed responses to the reviewers' comments are provided point by point below:
“The article reads as two parts: the first being a review of replication itself and the second being a discussion of techniques. Again this is somewhat inevitable, but the authors should try to ensure that the two parts integrate better.”
Thank you very much for all your comments and corrections. We have taken care to include them all in the manuscript. All changes to the text of the manuscript have been put in blue font.
We have created a short chapter that links the section on DNA replication with the section on methodological approaches. Our idea to integrate the content of the two parts was to include the points at the start of the description of each technique/method. These points indicate what questions the technique can answer, what data can be obtained with it or what aspect of the replication process it analyses. To better highlight the integrative purpose of these points, we have modified them slightly. In addition, the column entitled 'DNA replication stage' in Table 1 lists the stages of the DNA replication process that can be analyzed by a particular method or technique.
“On the whole the article is well-written, but there are quite a few minor errors throughout, and the article would benefit from being read by a native English speaker.”
The article has been red by native English speaker.
“I felt that more figures that exemplified the use of different methods would be helpful.”
Adding sample images for each technique would have made the paper better, but the review paper has been already quite long. We think adding pictures would have been too much, especially as each result should be explained what can be seen on it. This is especially necessary for techniques whose results can't be understood with basic knowledge of molecular methodology but require special interpretation. Also, we would have needed special permission to use photos of published results, and there would have been a lot of photos. Next to each technique, there are references to publications that used the technique and showed both the results and their interpretation.
“Specific comments in table”
Thank you very much for pointing out the errors in the text. We have carefully made the corrections.